# Insertion of an Amphipathic Linker in a Tetrapodal Tryptophan Derivative Leads to a Novel and Highly Potent Entry Inhibitor of Enterovirus A71 Clinical Isolates

**DOI:** 10.3390/ijms24043539

**Published:** 2023-02-10

**Authors:** Olaia Martí-Marí, Rana Abdelnabi, Dominique Schols, Johan Neyts, María-José Camarasa, Federico Gago, Ana San-Félix

**Affiliations:** 1Instituto de Química Médica (IQM, CSIC), E-28006 Madrid, Spain; 2Laboratory of Virology and Chemotherapy, Rega Institute for Medical Research, Department of Microbiology and Immunology, University of Leuven, B-3000 Leuven, Belgium; 3Departamento de Ciencias Biomédicas y Unidad Asociada IQM-UAH, Universidad de Alcalá, E-28805 Alcalá de Henares, Spain

**Keywords:** tryptophan, SAR, antiviral agents, HIV, EV71

## Abstract

**AL-471**, the leading exponent of a class of potent HIV and enterovirus A71 (EV-A71) entry inhibitors discovered in our research group, contains four l-tryptophan (Trp) units bearing an aromatic isophthalic acid directly attached to the C2 position of each indole ring. Starting from **AL-471**, we (i) replaced l-Trp with d-Trp, (ii) inserted a flexible linker between C2 and the isophthalic acid, and (iii) substituted a nonaromatic carboxylic acid for the terminal isophthalic acid. Truncated analogues lacking the Trp motif were also synthesized. Our findings indicate that the antiviral activity seems to be largely independent of the stereochemistry (l- or d-) of the Trp fragment and also that both the Trp unit and the distal isophthalic moiety are essential for antiviral activity. The most potent derivative, **23** (**AL-534**), with the C2 shortest alkyl urea linkage (three methylenes), showed subnanomolar potency against different EV-71 clinical isolates. This finding was only observed before with the early dendrimer prototype **AL-385** (12 l-Trp units) but remained unprecedented for the reduced-size prototype **AL-471**. Molecular modeling showed the feasibility of high-affinity binding of the novel l-Trp-decorated branches of **23** (**AL-534**) to an alternative site on the VP1 protein that harbors significant sequence variation among EV-71 strains.

## 1. Introduction

Acquired immune deficiency syndrome (AIDS), caused by human immunodeficiency virus type 1 (HIV-1), is a global health problem for which there is still not a curative treatment and no vaccine. HIV-1 is a retrovirus that attacks the immune system, resulting in vulnerability to opportunistic infections. HIV infection affected more than 37 million people worldwide and caused about 1 million deaths from AIDS-related causes in 2021 [1]. The introduction of highly active antiretroviral therapy (HAART), in the late 1990s, led to a dramatic reduction in the level of circulating virus, thus improving the quality and length of life of infected people who have access to medication [2,3,4]. Nonetheless, despite the remarkable success of HAART, issues such as viral heterogeneity, drug-associated toxicity, poor tolerability, and lack of adherence and compliance can lead to treatment failure and may result in the selection of viruses with mutations that confer resistance to one or more antiretroviral agents [5,6].

Current HAART usually targets reverse transcriptase, protease, and integrase because of the long-standing availability of biochemical assays for screening these viral enzymes [7]. However, any function that is essential for viral life may be considered as a potential target for antiviral drug development [8]. One such process is HIV entry into the host cell [9,10] during which the glycoprotein gp120 of the viral envelope plays a key role because it makes the first contact with the host cell CD4 receptors [11]. Because of this, gp120 is considered an important albeit underexploited therapeutic target nowadays [12].

Enterovirus 71 (EV-71) is a nonenveloped pathogenic (+)RNA virus (genus *Enterovirus*, family *Picornaviridae*) with pandemic potential able to provoke hand, foot, and mouth disease (HFMD), a mild syndrome that affects mostly children younger than 5 years old [13,14,15,16]. Unlike other HFMD-related enteroviruses, EV71 may also cause severe neurological/cardiopulmonary complications, leading to mortality. Currently, although three vaccines against EV-71 have been licensed in China, all using C4 genogroup strains, no vaccine or antiviral agent against EV-71 has been approved by the US FDA yet due to a series of concerns [17,18]. Therefore, the development of effective and specific antiviral drugs is urgently needed [19].

In previous work, we described the synthesis and biological activity of potent entry inhibitors of both HIV and EV-A71 [20,21]. The earlier compounds, whose prototype is **AL-385**, were negatively charged dendrimers with 9–12 L-tryptophan (Trp) units on the periphery (Figure 1). A scaffold simplification strategy led to a second generation of l-Trp-containing tripodal and tetrapodal derivatives [22]. Representative prototypes of this second family are the tripodal **AL-470** and tetrapodal **AL-471** compounds, which contain either three or four Trp residues, respectively, each bearing an isophthalic acid moiety directly attached to the C2 position of the indole ring.

Mechanistic studies revealed that **AL-470** and **AL-471**, as well as dendrimer **AL-385**, interact directly both with gp120 of HIV and with the fivefold axis of the EV-A71 capsid, in particular with VP1 residues Lys244 (K244) and Tyr 245 (Y245). Consequently, these compounds interfere with the binding of HIV/EV-A71 to their corresponding host cell surface receptors, thus preventing viral entry and cell infection [22,23].

To enrich the structure–activity relationship (SAR) studies of these promising series further, we here carried out modifications on the tetrapodal prototype **AL-471** by preparing compounds of general formula **I** and **II** (Figure 2). First, to assess the stereochemical preference of the Trp moiety for antiviral activity, a new tetrapodal derivative with d-Trp instead of l-Trp, but also bearing the isophthalic acid at C2, was synthesized. Second, the crucial isophthalic acid moiety attached to l-Trp was separated from the C2 position through a flexible linker of different length. Third, the aromatic isophthalic acid residue, at the end of the C2 alkyl chain, was replaced with a nonaromatic carboxylic acid. Finally, to shorten the synthetic route, we also prepared truncated analogues of general formula **II**, in which the Trp moiety was removed and only four isophthalic acid fragments were attached, either directly or through a linker, to the central tetrapodal scaffold (Figure 2).

## 2. Results and Discussion

### 2.1. Chemical Results 

The d-Trp isomer of **AL-471** was synthesized as outlined in Figure 1. Commercially available *N^α^*-Cbz-d-Trp methyl ester **1** was C2-arylated with dimethyl 5-iododisophthalate through a palladium-catalyzed C−H activation following Lavilla’s conditions (Figure 1) [24,25]. The reaction was carried out in anhydrous DMF, 5 mol% Pd(OAc)_2_, AgBF_4_, TFA, and microwave (MW) irradiation for 30 min at 120 °C to afford intermediate **2** in 55% yield. As previously described for l-Trp, the mild reaction conditions allowed the stereochemical integrity of the Trp unit to be preserved [22]. Subsequent carboxybenzyl (Cbz) deprotection through catalytic transfer hydrogenolysis [26] afforded the amino intermediate **3** in good yield (83%). Next, HATU-mediated coupling of **3** with the tetrapodal amino scaffold **4** [27] afforded intermediate **5** (74%). Finally, basic hydrolysis of the ester moieties, using LiOH·H_2_O and subsequent acidification with 1 M HCl, gave the fully deprotected compound **6** in 82% yield.

The synthetic route to **23**–**25**, in which the crucial isophthalic acid moiety is separated from the C2 position through a flexible linker of different length, is described in Figure 2. The synthesis started with the commercially available N^α^-Boc-l-Trp methyl ester (**7**), which was C2 alkylated with the corresponding *N*-(bromoalkyl)phthalimide following Batch’s conditions [28]. This method involves a Pd(II)-catalyzed norbornene-mediated C-H activation reaction in which norbornene acts as a cocatalyst that transposes the palladium metal from the initially attacked indole nitrogen atom to the C2 position of the indole ring. The subsequent addition of the primary alkyl bromide (N-(bromoalkyl)phthalimide), under basic conditions (K_2_CO_3_), gave the desired intermediates **8**–**10** in good yields (60–85%). Subsequent hydrazinolysis gave amines **11**–**13**, which were reacted with dimethyl 5-isocyanateisophthalate to afford ureas **14**–**16** (40–81%). Finally, Boc deprotection in acidic medium (TFA) gave intermediates **17**–**19**, which were used for the next step without purification. Subsequent HATU-mediated coupling of **17**–**19** with the commercially available tetrapodal scaffold **4** [27] at 30 °C afforded intermediates **20–22.** Finally, saponification of methyl ester groups (LiOH·H_2_O), followed by acidification (1 M HCl), gave the desired final tetrapodal compounds **23**–**25** (34–80%).

Additionally, compounds **32** and **33**, with a C2 alkyl chain bearing a terminal nonaromatic carboxylic acid, were synthesized as described in Figure 3. The synthesis started with the commercially available *N^α^*-Boc-l-Trp methyl ester (**7**), which was C2 alkylated with methyl 4-bromobutanoate and ethyl 7-bromoheptanoate using the norbornene-mediated C–H activation conditions [28] to afford intermediates **26** and **27** in high yields (96 and 77%, respectively). Selective NHBoc deprotection in acidic medium (TFA) afforded intermediates **28** and **29**, with a free amino group, that were used in the next step without purification. Further coupling of **28** and **29**, through HATU-mediated conditions, with the tetrapodal scaffold **4** [27] afforded intermediates **30** (26%) and **31** (48%), respectively. Finally, hydrolysis in basic medium of the methyl ester groups (LiOH·H_2_O), followed by acidification (1 M HCl), gave the desired final compounds **32** (66%) and **33** (28%), respectively.

Next, the synthesis of the truncated analogue **35**, with only four isophthalic acid moieties directly attached to the central pentaerythritol-based scaffold, was achieved in two steps (Figure 4). First, HATU-mediated coupling reaction of the tetrapodal core **4** [27] with dimethyl 5-aminobenzene-1,3-dicarboxylate afforded intermediate **34** (75%). Subsequent hydrolysis of the methyl ester groups (LiOH·H_2_O), followed by acidification (1 M HCl), gave the desired final compound **35** (78%). 

Finally, the synthesis of **40**, with a two-methylene alkyl chain between the central scaffold and each peripheral isophthalic acid moiety, was achieved in four steps (Figure 5). First, commercially available benzyl (2-aminoethyl)carbamate **36** was reacted with dimethyl 5-isocyanatoisophthalate to afford urea **37** in 54% yield. Subsequent carboxybenzyl (Cbz) deprotection through catalytic transfer hydrogenolysis, using dry ammonium formate and 10% Pd/C [26], afforded the amino intermediate **38**. Subsequent coupling to the tetrapodal central scaffold **4** [27] gave intermediate **39** in 34% yield. Hydrolysis, under basic conditions, of the methyl ester groups (LiOH·H_2_O), followed by acidification (1 M HCl), gave the desired final compound **40** in excellent yield**.**

### 2.2. Biological Results

#### 2.2.1. Antiviral Activity against HIV 

The newly synthesized compounds were first evaluated against HIV-1 and HIV-2 infections in cultured CD4^+^ T cells (Table 1). Activity profiling was based on the inhibition of the HIV-induced cytopathic effect on MT-4 cells following described procedures [29]. The early prototypes **AL-471** and **AL-385**, together with dextran sulfate-5000 (DS-5000), a negatively charged HIV adsorption inhibitor [30], and pradimicin A (PRM-A), a gp120 carbohydrate-binding entry inhibitor [31], were selected as reference compounds. These positive controls showed antiviral activities within previously reported ranges [30,31]. 

As shown in Table 1, all compounds, except **35** and **40**, which resulted as inactive at 50 µM, exhibited anti-HIV-1 activity in the (sub)micromolar range, with EC_50_ values ranging from 16 µM (**33**) to 0.03 µM (**6**). Interestingly, the activity of the d-Trp analogue **6** (EC_50_: 0.03 µM, HIV-1; 0.02 µM, HIV-2) did not change significantly with respect to that of the l-Trp containing prototype **AL-471** (EC_50_: 0.07 µM, HIV-1; 0.02 µM, HIV-2). This finding suggests that the stereochemistry of the amino acid is not crucial for the anti-HIV activity.

The effect of the alkyl chain length on anti-HIV activity was analyzed with compounds **23–25** (Table 1). The greatest anti-HIV potency (EC_50_: 0.12 µM) was observed for the C2 alkyl derivative **23**, with an alkyl chain containing only three methylenes. Further length extension of the aliphatic chain yielded the less potent **24** (five methylenes, EC_50_: 0.43 µM) and **25** (eight methylenes, EC_50_: 3.41 µM). Overall, the fact that **23–25** were significantly less potent (EC_50_: 0.12–3.41 µM) than the prototype **AL-471** (EC_50_: 0.07 µM) indicates that the presence of the aliphatic chain is detrimental for anti-HIV-1 activity. In contrast, with respect to HIV-2, **23** and **24** (EC_50_: 0.02 and 0.03 µM, respectively) were as potent as the prototype **AL-471** (EC_50_: 0.02 µM), whereas **25** (EC_50_: 1.40 µM) clearly lost activity. 

Substitution of the aromatic isophthalic acid at the end of the C2 alkyl chain with a nonaromatic carboxylic acid led to a reduction in potency, as exemplified by **32** and **33** (EC_50_: 5.7, 16 µM), which were much less potent than the isophthalic acid derivatives **23–25** (EC_50_: 0.12–3.4 µM).

Finally, the Trp-less **35**, with four isophthalic acid moieties directly attached to the central scaffold, and **40**, with a spacer linking the four isophthalic acid residues to the central scaffold, were not active against HIV-1 at concentrations up to 50 µM. However, **40** showed significant activity against HIV-2 (EC_50_: 4.83 µM). 

#### 2.2.2. Antiviral Activity against EV-71 

The newly synthesized compounds were also evaluated against enterovirus A71 (BrCr strain) infection in cultured rhabdomyosarcoma (RD) cells, which are known for their high susceptibility to EV-A71-induced cell death [32]. The previously described prototypes **AL-471** and **AL-385**, together with pirodavir, an entry inhibitor that interacts with the viral capsid, were also assayed under the same conditions for comparative purposes (Table 2) [21,22,33,34].

As observed for HIV, all the synthesized compounds exhibited anti-EV-A71 activities in the (sub)micromolar range, with EC_50_ values ranging from 17 µM (**40**) to 0.02 µM (**23**, **AL-534**). Compound **6** (EC_50_: 0.10 µM), the d-counterpart of the prototype **AL-471** (EC_50_: 0.04 µM), turned out to be only twofold less potent. Therefore, the stereochemistry of the amino acid seems not to be crucial for the anti-EV-A71 activity. 

Within the C2 alkylated isophthaloyl derivatives (**23–25**), the most potent was **23** (**AL-534**) (EC_50_: 0.02 µM), which bears the shortest alkyl chain (three methylenes), followed by **24** (EC_50_: 0.29 µM), and **25** (EC_50_: 2.86 µM), with alkyl chains of five and eight methylenes, respectively. Compound **23** (**AL-534**) was twofold (EC_50_: 0.02 µM) more potent than the prototype **AL-471** (EC_50_: 0.04 µM), whereas **24** and **25** were less potent. These findings attest to the importance of alkyl chain length for anti-EV-A71 activity. 

Compounds **32** (EC_50_: 4.69 µM) and **33** (EC_50_: 7.29 µM), bearing an alkyl chain at C2 with a monocarboxylic acid at the end, were considerably less potent than the isophthalic acid derivatives **23–25** (EC_50_: 0.13–2.86 µM). These data reinforce the important role of the isophthalic acid moiety for anti-EV-A71 activity.

Finally, **35** and **40**, in which four isophthalic acid residues (no Trp motif) were linked directly or through a spacer to the tetrapodal central scaffold, were considerably less potent (EC_50_: 8.5 and 17.1 µM, respectively) than the prototype **AL-471** (EC_50_: 0.04 µM), thus confirming that the presence of the Trp unit is important for activity. 

In summary, **23** (**AL-534**), with the shortest alkyl chain (three methylenes) between the C2 position of the indole ring and the distal isophthalic acid in each pendant leg, proved to be the most potent of this series against EV-A71, and for this reason it was selected for additional biological and structural studies. 

Virus-cell-based assays using **23** (**AL-534**) against a panel of representative enteroviruses (Table 3) showed no activity on coxsackievirus B3 and rhinoviruses A and B, and only some activity against enterovirus D68 (EC_50_: 26 µM). Taken together, these results indicate that **23** (**AL-534**) is quite a specific inhibitor of EV-A71 infection. 

Furthermore, when **23** (**AL-534**) was tested against a panel of clinical isolates representative of the different EV-71 (sub)genogroups (B2, B5, C2, and C4) and compared to **AL-471** and **AL-385** (Table 4), we found that its activity was even greater than that found for the lab-adapted BrCr strain (subgenogroup A). In particular, the greatest enhancement was observed against subgenogroups B5 and C4 (90- and 45-fold improvement, respectively, with respect to the BrCr strain), whereas the lowest enhancement was detected for subgenogroups B2 and C2 (ninefold improvement). This subnanomolar potency against the clinical isolates B5 and C4 (EC_50_: 0.2 and 0.4 nM) was observed earlier only with the dendrimer prototype **AL-385** (EC_50_: 0.2 nM in both cases) but not with the rest of reduced-sized compounds described until now, out of which **AL-471** (EC_50_: 5.9 and 1.4 nM) was the most active member [21,22].

Finally, evaluation of **23** (**AL-534**) against single-mutant (VP1_S184T and VP1_P246S) and double-mutant (VP1_S184T_P246S) EV-A71 strains that are resistant to **AL-385** showed diminished antiviral activity (EC_50_: 0.41–0.58 µM vs. 0.02 µM for the wild-type virus) (Table 5). Since **AL-385** has been shown to bind to protein VP1, which makes up the fivefold vertex of the viral capsid [23], this result suggests that the binding site for **23** (**AL-534**) is likely to overlap—at least partially—with that of **AL-385**. To find out why the binding affinity of **23** (**AL-534**) is still high enough in the presence of the S184T and P246S amino acid substitutions, we carried out some molecular modeling studies. 

### 2.3. Molecular Modeling Studies 

Visual inspection of the models previously generated for **AL-471** bound to the fivefold axis of the EV-A71 capsid protein VP1 [22] suggests that replacement of l-Trp units with d-Trp would generate similar binding modes for **6** (Appendix A). With respect to HIV, the detailed models reporting the interaction of a membrane-embedded and heavily glycosylated HIV-1 Env trimer with the doubly arylated tetrapodal derivative **AL-518**—previously described by our group [35]—revealed preferential binding to a location close to the V3 loop of gp120 that can be similarly targeted by the new **23** (**AL-534**). Subtle differences in potency against HIV-1 and HIV-2 for **23** (**AL-534**) are to be expected due to (i) well-known sequence and loop differences between HIV-1 and HIV-2, (ii) the multiple patterns of simultaneous CH···π stacking and hydrogen-bonding interactions that are possible between the distinct isophthalic acid-decorated Trp units and the oligomannose glycans attached to Asn residues at positions 134, 197, and 363 in HIV-1, (iii) modulation of electrostatic interactions with the side-chain guanidinium moieties of Arg151 (conserved in HIV-2) and Arg469 (Thr481 in HIV-2) [35], and (iv) the fact that antibodies do not usually cross-react between HIV-1 and HIV-2. Studying this variation in depth is further complicated by the fact that Env proteins gp120 and gp41 subunits in the different HIV-1/HIV-2 subtypes or clades exist as populations of glycosylated variants (glycoforms) that are characterized by heterogeneous patches of oligomannose-type glycans at each N-linked glycosylation site.

For the abovementioned reasons, we focused our molecular modeling efforts on the binding of **23** (**AL-534**) to the fivefold axis of the EV-A71 capsid protein VP1 (Figure 3A) in an attempt to account for its superior activity on B5 and C4 clinical isolates over the lab-adapted BrCr strain. Our docking results using one of the decorated l-Trp pendant “legs” of **23** (**AL-534**) as the ligand provided alternative suitable locations, of which that showing the largest clustering and highest score (−7.2 kcal mol^−1^) was found at the interface between neighboring subunits (Figure 3B). This high-affinity binding pose (Figure 3C) features (i) the Trp indole ring prominently lodged over the hydrophobic surface provided by Pro96, Pro142, and Pro246, (ii) the urea moiety of the sandwiched linker with its dipole moment properly oriented between the side-chain hydroxyl of Ser240 on one side and the carboxylates of Glu92 and Asp94 on the other side, and (iii) the isophthalic acid carboxylates engaged in favorable electrostatic interactions with Arg236 from one VP1 subunit and Arg250 from the neighboring subunit.

Having determined the feasibility of this binding mode by the structural stability observed during the ensuing MD simulation of the complex of a VP1 pentamer with five identical molecular fragments (Appendix A), we then modeled and simulated the corresponding complex with a whole **23** (**AL-534**) molecule bound at the fivefold vertex. This allowed us to see that while the decorated l-Trp unit in one of the “legs” maintained multiple specific interactions with selected residues at the interface between two VP1 subunits, as described above, the other three engaged in additional contacts with neighboring subunits and amino acids lining the pore by exploiting the concept of multivalency [36]. This binding mode (Figure 4) should prevent interaction of the virus with attachment receptors such as SCARB2, Tyr-sulfated PSGL-1, heparan sulfate proteoglycans, and sialylated glycans on the host cell membrane [37,38] so that EV-A71 entry into the host cells is effectively blocked.

## 3. Materials and Methods

### 3.1. Synthesis

*General Chemical Procedures*. Commercial reagents and solvents were used as received from the suppliers without further purification unless otherwise stated. The solvents employed in some reactions were dried prior to use. DMF dry was commercially available (Aldrich, St. Louis, MO, USA).

A microwave reactor Emrys^TM^ Synthesizer (Biotage AB, Uppsala, Sweden) was used for the reactions which needed microwave irradiation (synthesis of intermediate **2**).

Analytical thin-layer chromatography (TLC) was performed on aluminum plates precoated with silica gel 60 (F_254_, 0.20 mm). Products were visualized using an ultraviolet lamp (254 nm and 365 nm) or by heating after treatment with a 5% solution of phosphomolybdic acid (PMA) or vanillin in ethanol. 

The compounds were purified by (a) high-performance flash chromatography (HPFC) with an “Isolera One” (Biotage) system in reverse phase using water/acetonitrile (100:0 to 0:100) as eluent, (b) flash column chromatography on silica gel (60 Merck 230–400 mesh), (c) preparative centrifugal circular thin layer chromatography (CCTLC) on a Chromatotron^®^ (Kiesegel 60 PF254 gipshaltig, Merck, Rahway, NJ, USA) with layer thickness 1 mm and a flow rate of 2–4 mL/min. 

For HPLC analysis, an Agilent Technologies 1120 Compact LC with a reverse-phase column ACE 5 C18-300 (4.6 mm × 150 mm, 3.5 μm) equipped with a PDA (photo diode array) detector was used. Acetonitrile was used as mobile phase A, and water with 0.05% of TFA was used as mobile phase B with at a flow rate of 1 mL·min^−1^. All retention times are quoted in minutes and the gradients are specified for each compound in the experimental data.

For high-resolution mass spectrometry (HRMS), an Agilent 6520 Accurate Mass QTOF (quadrupole time of flight) coupled with LC/MS and equipped with an electrospray interface (ESI) working in the positive-ion (ESI^+^) and negative-ion (ESI^−^) mode was used.

NMR spectra (^1^H, ^13^C NMR) were recorded on Varian UNIT INOVA-300 (300 MHz), Bruker AVANCE 300 (300 and 75 MHz), Varian INOVA-400 (400 and 100 MHz), Varian MERCURY-400 (400 and 100 MHz), and Varian-500 (500 and 125 MHz) spectrometers, using DMSO-*d_6_* or CDCl_3_ as solvents. Chemical shift (*δ*) values are reported in parts per million (ppm) relative to tetramethylsilane (TMS) in ^1^H and CDCl_3_ (*δ* = 77.0) in ^13^C NMR. Coupling constant (*J* values) are reported in hertz (Hz), and multiplicities of signals are indicated by the following symbol: s (singlet), d (doublet), t (triplet), q (quadruplet), m (multiplet), and bs (broad singlet). Some two-dimensional spectra (COSY, HSQC, and HMBC) were performed to identify the structure. 

The values of specific rotation were determined in a Perkin Elmer Polarimeter (Model 141).

The final compounds were lyophilized using a Telstar 6–80 system. Their purity was at least 95% based on HPLC, LC/MS, and ^1^HNMR analyses. All compounds are >95% pure by HPLC analysis. 

#### 3.1.1. General Procedure for C2 Alkylation of Trp Derivatives

To a mixture of commercially available *N^α^-*Boc-*L*-tryptophan methyl ester **7** (0.50 mmol, 1.00 equiv), norbornene (1.00 mmol, 2.00 equiv), K_2_CO_3_ (3.00 mmol, 4.00 equiv), and PdCl_2_ (10 mol%), 2.5 mL of DMF (containing 0.5 M H_2_O) and the corresponding primary alkyl bromide (2.00 mmol, 4.00 equiv) were added. The resulting suspension was stirred at 60 °C for 24 h. After being cooled to room temperature, the reaction mixture was concentrated under reduce pressure and then diluted with EtOAc (20 mL) and washed with water (3 × 20 mL). The combined organic extracts were dried over anhydrous Na_2_SO_4_ and concentrated. The crude product was purified by flash column chromatography on silica gel or CCTLC to afford the corresponding 2-alkyltryptophan intermediate. 

#### 3.1.2. General Coupling Procedure for the Synthesis of OMe-Protected Tetrapodal Trp Derivatives

To a solution of the tetrapodal polyacid **4** [27] (1.00 equiv), in anhydrous DMF (10 mL), HATU (1.20 equiv each carboxylic acid group), the corresponding OMe protected C-2 alkyl tryptophan derivative (1.20 equiv each carboxylic acid group), and DIPEA (2.40 equiv each carboxylic acid group) were added. The reaction mixture was stirred under argon atmosphere at 30 °C for 12 h and then evaporated to dryness. The residue was dissolved in ethyl acetate (20 mL) and washed successively with aqueous solutions of citric acid (10%) (3 × 20 mL), saturated NaHCO_3_ (3 × 20 mL), and brine (3 × 20 mL). The organic phase was dried over anhydrous Na_2_SO_4_, filtered, and evaporated to dryness. The crude product was purified by Biotage HPFC (high-performance flash chromatography) purification system on reverse phase using water/acetonitrile (100:0 to 0:100) as eluent to afford the corresponding products.

#### 3.1.3. General Procedure for Methyl Ester Deprotection

To a solution containing the corresponding methyl ester derivative (1.00 equiv) in THF (10 mL) at 0 °C (ice bath), a solution of LiOH∙H_2_O (2.00 equiv for each methyl ester group) in water (2 mL) was added, and the mixture was stirred at 30 °C overnight. Then 1 M hydrochloric acid aqueous solution was added to reach pH~2, and volatiles were evaporated to dryness. The residue was dissolved in *iso-*butanol (15 mL) and washed with brine (3 × 10 mL) and water (3 × 10 mL). The organic phase was dried over anhydrous Na_2_SO_4_, filtered, and evaporated to dryness. The residue was purified with a Biotage HPFC (high-performance flash chromatography) purification system on reverse phase using water/acetonitrile (100:0 to 0:100) as eluent, frozen and lyophilized, yielding the product as a fluffy powder. 

#### 3.1.4. Dimethyl (*R*)-5-[3-(2-{[(Benzyloxy)carbonyl]amino}-3-methoxy-3-oxopropyl)-1*H*-indol-2-yl]isophthalate **2**


Commercially available *N^α^*-benzyloxycarbonyl-d-tryptophan methyl ester **1** (680 mg, 1.93 mmol, 1.00 equiv), dimethyl 5-iodoisophthalate (927 mg, 2.90 mmol, 1.50 equiv), Pd(OAc)_2_ (22 mg, 0.10 mmol, 5 mol%), AgBF_4_ (751 mg, 3.86 mmol, 2.00 equiv), and TFA (148 μL, 1.93 mmol, 1.00 equiv) were placed in an MW reactor vessel under argon atmosphere in anhydrous DMF (12 mL). The mixture was heated under MW irradiation (250 W) at 120 °C for 30 min. The resulting suspension was filtered through Whatman^®^ filter paper 42, and the solvent was removed under vacuum. The residue was dissolved in ethyl acetate (20 mL) and washed successively with saturated NaHCO_3_ (3 × 20 mL) and brine (3 × 20 mL). The organic phase was dried over anhydrous Na_2_SO_4_, filtered, and evaporated to dryness. The residue was purified by flash column chromatography using hexane:ethyl acetate (7:3) to afford **2** (575 mg, 55%) as an amorphous solid of cream color. ^1^H NMR (300 MHz, CDCl_3_) *δ*: 8.59 (p, *J* = 1.6 Hz, 1H, NH-1^i^Trp), 8.34 (d, *J* = 7.5 Hz, 3H, Ar), 7.59 (d, *J* = 7.9 Hz, 1H, Ar), 7.38–7.18 (m, 5H, Ar), 7.10 (t, *J* = 7.3 Hz, 1H, Ar), 5.16 (d, *J* = 8.3 Hz, 1H, NHCO), 4.97–4.76 (m, CH_2_Ph), 4.70–4.55 (m, 1H, *α*-CHTrp), 3.91 (s, 6H, OCH_3_), 3.46 (m, 2H, *β*-CH_2_Trp), 3.36 (s, 3H, OCH_3_). HPLC (gradient: H_2_O:MeCN, 10–100% of MeCN in 10 min): 9.327 min. 

#### 3.1.5. Dimethyl (*R*)-5-[3-(2-Amino-3-methoxy-3-oxopropyl)-1*H-*indol-2-yl]isophthalate **3**

To a solution of compound **2** (570 mg, 1.05 mmol, 1.00 equiv) in DMF (10 mL), ammonium formate (219 mg, 3.45 mmol, 3.30 equiv) and Pd/C (10% on C; 30 wt %) were added. After 3 h under argon atmosphere, the residue was filtered through a Whatman^®^ filter paper 42 and the solvent was removed under reduced pressure to give 356 mg (83%) of **3** as an amorphous white solid. The crude was used in the next step without purification. HPLC (gradient: H_2_O:MeCN, 10 min: 6.277 min.

#### 3.1.6. Tetramer **5**


According to the general coupling procedure, compound **3** (340 mg, 0.83 mmol, 4.80 equiv), was added to a mixture of **4** [27] (73 mg, 0.17 mmol, 1.00 equiv), HATU (315 mg, 0.83 mmol, 4.80 equiv), and DIPEA (281 µL, 1.73 mmol, 10.00 equiv). Purification by Biotage HPFC (high-performance flash chromatography) system on reverse phase using water:acetonitrile (100:0 to 0:100) yielded **5** (254 mg, 74%). ^1^H NMR (400 MHz, DMSO-*d_6_*) *δ*: 11.52 (s, 3H, NH-1^i^Trp), 8.44 (t, *J* = 1.6 Hz, 3H, Ar), 8.39 (d, *J* = 1.7 Hz, 8H, Ar), 8.32 (d, *J* = 7.7 Hz, 4H, NHCO), 7.59 (d, *J* = 7.5 Hz, 4H, Ar), 7.37 (d, *J* = 7.4 Hz, 4H, Ar), 7.13 (t, *J* = 7.5 Hz, 4H, Ar), 7.01 (t, *J* = 7.4 Hz, 4H, Ar), 4.62–4.48 (m, 4H, *α*-CHTrp), 3.90 (s, 24H, OCH_3_), 3.38–3.29 (m, 26H, OCH_3_ + *β*-CH_2_Trp + CH_2_), 3.20 (dd, *J* = 14.5, 6.6 Hz, 4H, *β*-CH_2_Trp), 3.06 (m, 8H, CH_2_), 2.20 (m, 4H, CH_2_), 2.16–2.02 (m, 4H, CH_2_). HPLC (gradient: H_2_O:MeCN, 50–100% of MeCN in 10 min): 7.682 min.

#### 3.1.7. Tetramer **6**


According to the general procedure for deprotection of methyl esters, compound **5** (100 mg, 0.05 mmol, 1.00 equiv) was treated with LiOH·H_2_O (51 mg, 1.20 mmol, 24.00 equiv). Purification by Biotage HPFC (high-performance flash chromatography) system on reverse phase using water/acetonitrile (100:0 to 0:100) yielded **6** (75 mg, 82%). ^1^H NMR (500 MHz, DMSO-*d_6_*) *δ*: 11.43 (s, 4H, NH-1^i^Trp), 8.45 (t, *J* = 1.6 Hz, 4H, Ar), 8.41–8.32 (m, 8H, Ar), 8.18 (d, *J* = 8.0 Hz, 4H, NHCO), 7.66 (d, *J* = 8.0 Hz, 4H, Ar), 7.35 (d, *J* = 8.1 Hz, 4H, Ar), 7.11 (t, *J* = 8.1, 4H, Ar), 7.03–6.96 (m, 4H, Ar), 4.66–4.33 (m, 4H, *α*-CHTrp), 3.40–3.33 (m, 4H, *β*-CH_2_Trp), 3.29 (m, 8H, CH_2_), 3.17 (dd, *J* = 14.3, 6.9 Hz, 4H, *β*-CH_2_Trp), 3.08 (s, 8H, CH_2_), 2.21 (dt, *J* = 13.9, 6.7 Hz, 4H, CH_2_), 2.11 (dq, *J* = 16.1, 9.5, 8.0 Hz, 4H, CH_2_). ^13^C NMR (126 MHz, DMSO-*d_6_*) *δ*: 173.0, 169.9, 166.6, 136.2, 133.7, 133.6, 132.6, 132.0, 128.6, 128.6, 122.0, 119.2, 119.0, 111.3, 108.5, 68.8, 67.0, 52.8, 44.8, 35.6, 27.2. HPLC (gradient: H_2_O:MeCN, 10–100% of MeCN in 10 min): 6.814 min. [α]_D_ = −15.50° (c1, MeOH). The specific rotation observed for **6** is opposite in sign to those found for the prototype **AL-471**, [α]_D_ = +16.10° (c1, MeOH), confirming that both compounds are enantiomers. 

#### 3.1.8. Methyl 2-[(*tert*-Butoxycarbonyl)amino]-3-{2-[3-(1,3-dioxoisoindolin-2-yl)propyl]-1*H*-indol-3-yl}propanoate **8**


According to the general procedure for C2 alkylation, commercially available *N^α^*-Boc-(*S*)-tryptophan methyl ester **7** (800 mg, 2.51 mmol, 1.00 equiv) was treated with norbornene (473 mg, 5.03 mmol, 2.00 equiv), K_2_CO_3_ (1.78 g, 10.05 mmol, 4.00 equiv), PdCl_2_ (35 mg, 0.25 mmol, 10 mol%), and *N-(*3-bromopropyl)phthalimide (2.70 g, 10.05 mmol, 4.00 equiv). Purification by flash column chromatography using hexane:ethyl acetate (6:4) yielded **8** (744 mg, 60%) as a pale-yellow oil. ^1^H NMR (400 MHz, CDCl_3_) *δ*: 9.09 (s, 1H, NH-1^i^Trp), 7.83 (m, 2H, Ar), 7.71 (m, 2H, Ar), 7.43 (d, *J* = 7.9 Hz, 1H, Ar), 7.34 (d, *J* = 7.9 Hz, 1H, Ar), 7.10 (t, *J* = 7.5 Hz, 1H, Ar), 7.04 (t, *J* = 7.5 Hz, 1H, Ar), 5.06 (d, *J* = 8.5 Hz, 1H, NHCO), 4.58 (m, 1H, α-CHTrp), 3.78 (t, *J* = 6.3 Hz, 2H, CH_2_), 3.62 (s, 3H, OCH_3_), 3.21 (m, 2H, *β*-CH_2_Trp), 2.74 (t, *J* = 8.0 Hz, 2H, CH_2_), 2.04 (m, 2H, CH_2_), 1.39 (s, 9H, CH_3_). ^13^C NMR (101 MHz, CDCl_3_) *δ*: 173.0, 169.1, 155.2, 136.0, 135.5, 134.3, 132.0, 128.8, 123.5, 121.5, 119.5, 118.3, 110.8, 105.9, 79.9, 54.3, 52.4, 37.3, 28.7, 28.4, 27.5, 22.6. HPLC (gradient: H_2_O:MeCN, 10–100% of MeCN in 10 min): 9.472 min. 

#### 3.1.9. Methyl 2-[(*tert*-Butoxycarbonyl)amino[-3-{2-[5-(1,3-dioxoisoindolin-2-yl)pentyl]-1*H*-indol-3-yl}propanoate **9**


According to the general procedure for C2 alkylation, *N^α^*-Boc-(*S*)-tryptophan methyl ester **7** (800 mg, 2.51 mmol, 1.00 equiv) was treated with norbornene (473 mg, 5.03 mmol, 2.00 equiv), K_2_CO_3_ (1.78 mg, 10.05 mmol, 4.00 equiv), PdCl_2_ (35 mg, 0.25 mmol, 10 mol%), and *N*-(5-bromopentyl)phthalimide (3.00 g, 10.05 mmol, 4.00 equiv). Purification by flash column chromatography using hexane:ethyl acetate (6:4) yielded **9** (1.19 g, 85%) as a pale-yellow oil. ^1^H NMR (300 MHz, CDCl_3_) *δ*: 8.13 (s, 1H, NH-1^i^Trp), 7.82 (m, 2H, Ar), 7.70 (m, *2*H, Ar), 7.43 (d, *J* = 7.5 Hz, 1H, Ar), 7.29 (d, *J* = 7.5 Hz, 1H, Ar), 7.08 (m, 2H, Ar), 5.06 (d, *J* = 8.3 Hz, 1H, NHCO), 4.56 (m, 1H, α-CHTrp), 3.70 (m, 2H, NCH_2_), 3.61 (s, 3H, OCH_3_), 3.20 (d, *J* = 5.8 Hz, 2H, *β*-CH_2_Trp), 2.72 (t, *J* = 7.6 Hz, 2H, CH_2_), 1.74 (m, 4H, CH_2_), 1.41 (m, 11H, CH_2_ + CH_3_). ^13^C NMR (75 MHz, CDCl_3_) *δ*: 173.0, 168.7, 155.2, 137.0, 135.5, 134.1, 132.2, 128.8, 123.4, 119.5, 118.3, 110.5, 105.8, 79.8, 54.3, 52.4, 37.5, 28.7, 28.4, 28.3, 27.4, 26.3, 25.8. HPLC (gradient: H_2_O:MeCN, 40–100% of MeCN in 10 min): 8.287 min. HRMS (ESI^+^) *m/z*: calculated for C_30_H_35_N_3_O_6_ 533.25259; found 533.25164. 

#### 3.1.10. Methyl 2-[(*tert*-Butoxycarbonyl)amino]-3-{2-[8-(1,3-dioxoisoindolin-2-yl)octyl]-1*H*-indol-3-yl}propanoate **10**


According to the general procedure for C2 alkylation, *N^α^*-Boc-(*S*)-tryptophan methyl ester **7** (500 mg, 1.57 mmol, 1.00 equiv) was treated with norbornene (296 mg, 3.14 mmol, 2.00 equiv), K_2_CO_3_ (1.11 g, 6.28 mmol, 4.00 equiv), PdCl_2_ (22 mg, 0.16 mmol, 10 mol%), and *N*-(8-bromooctyl)phthalimide (2.12 g, 6.28 mmol, 4.00 equiv). Purification by flash column chromatography using hexane:ethyl acetate (6:4) yielded **10** (723 mg, 80%) as a pale-yellow oil. ^1^H NMR (300 MHz, CDCl_3_) *δ*: 8.14 (s, 1H, NH-1^i^Trp), 7.83 (m, 2H, Ar), 7.70 (m, 2H, Ar), 7.45 (d, *J* = 7.5 Hz, 1H, Ar), 7.26 (m, 1H, Ar), 7.08 (m, 2H, Ar), 5.05 (d, *J* = 8.3 Hz, 1H, NHCO), 4.58 (m, 1H, α-CHTrp), 3.68 (m, 2H, CH_2_), 3.63 (s, 3H, OCH_3_), 3.21 (d, *J* = 5.7 Hz, 2H, *β*-CH_2_Trp), 2.68 (m, 2H, CH_2_), 1.65 (m, 4H, CH_2_), 1.40 (s, 9H, CH_3_), 1.33 (s, 8H, CH_2_). ^13^C NMR (75 MHz, CDCl_3_) *δ*: 173.0, 168.7, 155.2, 137.6, 135.4, 134.0, 134.0, 132.3, 128.8, 123.3, 121.3, 119.5, 118.3, 110.4, 105.5, 79.8, 54.3, 52.3, 38.0, 29.6, 29.3, 29.2, 28.8, 28.6, 28.4, 27.4, 26.7, 26.1. HPLC (gradient: H_2_O:MeCN, 40–100% of MeCN in 10 min): 9.640 min. HRMS (ESI^+^) *m/z*: calculated for C_33_H_41_N_3_O_6_ 575.29954; found 575.30022. 

#### 3.1.11. Dimethyl 5-{[3-(3-{2-[(*tert*-Butoxycarbonyl)amino]-3-methoxy-3-oxopropyl}-1*H*-indol-2-yl)propyl]amino}isophthalate **14**

To a solution of compound **8** (24 mg, 0.11 mmol, 1.00 equiv) in EtOH (5 mL), hydrazine monohydrate (64%) (14 µL, 0.21 mmol, 2.00 equiv) was added. The resulting solution was stirred at 30 °C for 5 h. After being cooled to room temperature, the reaction mixture was concentrated under reduced pressure and then diluted with H_2_O (10 mL) and extracted with dichloromethane (3 × 15 mL). The combined organic extracts were washed with brine (3 × 15 mL), dried over anhydrous Na_2_SO_4_, and concentrated. The crude product **11** (30 mg, 0.08 mmol, 1.00 equiv) was dissolved under argon atmosphere in anhydrous DMF (5 mL), and dimethyl 5-isocyanatoisophthalate (23 mg, 0.10 mmol, 1.20 equiv) and DIPEA (16 µL, 0.10 mmol, 1.20 equiv) were added. The reaction mixture was stirred at 30 °C for 4 h and the crude product was purified by flash column chromatography on silica gel using hexane:ethyl acetate (3:7) to afford **14** (36 mg, 74%) as a pale-yellow oil. ^1^H NMR (400 MHz, CDCl_3_) *δ*: 9.07 (s, 1H, NH-1^i^Trp), 8.25 (t, *J* = 1.5 Hz, 1H, Ar), 8.18 (s, 2H, Ar), 7.66 (s, 1H), 7.41 (d, *J* = 6.2 Hz, 1H, Ar), 7.23 (m, 1H, Ar), 7.04 (m, 2H, Ar), 5.36 (d, *J* = 8.3 Hz, 1H, NHCO), 4.44 (m, 1H, *α*-CHTrp), 3.88 (s, 7H, CH_3_ + CH_2_), 3.68 (m, 1H, CH_2_) 3.57 (m, 3H, OCH_3_), 3.41 (m, 1H, *β*-CH_2_Trp), 3.22 (m, 3H, *β*-CH_2_Trp + CH_2_), 2.81 (m, 2H, CH_2_), 1.92 (m, 2H, CH_2_), 1.38 (s, 9H, CH_3_). ^13^C NMR (101 MHz, CDCl_3_) *δ*: 173.0, 166.4, 156.2, 155.8, 140.1, 136.5, 135.6, 131.1, 128.7, 124.5, 124.2, 121.3, 119.4, 117.8, 110.8, 105.7, 55.3, 52.6, 52.5, 39.2, 29.6, 28.4, 23.0. HPLC (gradient: H_2_O:MeCN, 10–100% of MeCN in 10 min): 9.305 min. 

#### 3.1.12. Dimethyl 5-{3-[5-(3-{2-[(*tert*-Butoxycarbonyl)amino]-3-methoxy-3-oxopropyl}-1*H*-indol-2-yl)pentyl]ureido}isophthalate **15**


To a solution of compound **9** (463 mg, 2.13 mmol, 1.00 equiv) in EtOH (5 mL), hydrazine monohydrate (64%) (158 µL, 2.09 mmol, 2.50 equiv) was added. The resulting solution was stirred at 30 °C for 5 h. After being cooled to room temperature, the reaction mixture was concentrated under reduced pressure and then diluted with H_2_O (10 mL) and extracted with dichloromethane (3 × 15 mL). The combined organic extracts were washed with brine (3 × 15 mL), dried over anhydrous Na_2_SO_4_, and concentrated. The crude product **12** (230 mg, 0.57 mmol, 1.00 equiv) was dissolved in anhydrous DMF (5 mL), and dimethyl 5-isocyanatoisophthalate (161 mg, 0.68 mmol, 1.20 equiv) and DIPEA (111 µL, 0.68 mmol, 1.20 equiv) were added. The reaction was stirred at 30 °C for 4 h and the crude product was purified by flash column chromatography on silica gel using hexane:ethyl acetate (6:4) to afford **15** (151 mg, 40%) as a pale-yellow oil. ^1^H NMR (300 MHz, CDCl_3_) *δ*: 8.75 (s, 1H, NH-1^i^Trp), 8.22 (s, 1H, Ar), 8.20 (s, 2H, Ar), 7.86 (m, 1H, NHCO), 7.40 (d, *J* = 7.2 Hz, 1H, Ar), 7.26 (m, 1H, Ar), 7.02 (m, 2H, Ar), 5.79 (s, 1H, NHCO), 5.20 (d, *J* = 8.1 Hz, 1H, NHCO), 4.50 (m, 1H, α-CHTrp), 3.82 (s, 8H, OCH_3_ + CH_2_), 3.57 (s, 3H, OCH_3_), 3.17 (m, 4H, *β*-CH_2_Trp + CH_2_), 2.63 (m, 2H, CH_2_), 1.62 (m, 2H, CH_2_), 1.36 (s, 11H, CH_3_ + CH_2_). ^13^C NMR (75 MHz, CDCl_3_) *δ*: 173.1, 166.3, 156.0, 155.6, 140.3, 137.3, 135.5, 131.1, 128.6, 124.3, 124.0, 121.1, 119.3, 117.9, 110.7, 105.2, 80.2, 54.8, 52.4, 39.7, 29.0, 28.4, 28.1, 26.5, 25.9. HPLC (gradient: H_2_O:MeCN, 10–100% of MeCN in 10 min): 9.343 min.

#### 3.1.13. Dimethyl 5-{3-[8-(3-{2-[(*tert*-Butoxycarbonyl)amino]-3-methoxy-3-oxopropyl}-1*H*-indol-2-yl)octyl]ureido}isophthalate **16**

To a solution of compound **10** (700 mg, 1.22 mmol, 1.00 equiv) in EtOH (5 mL), hydrazine monohydrate (64%) (178 µL, 3.65 mmol, 3.00 equiv) was added. The resulting solution was stirred at 30 °C for 5 h. After being cooled to room temperature, the reaction mixture was concentrated under reduced pressure and then diluted with H_2_O (10 mL) and extracted with dichloromethane (3 × 15 mL). The organic layers were washed with brine (3 × 15 mL), dried over anhydrous Na_2_SO_4_, and concentrated. The crude product **13** (500 mg, 1.12 mmol, 1.00 equiv) was dissolved in anhydrous DMF (5 mL), and dimethyl 5-isocyanatoisophthalate (317 mg, 1.35 mmol, 1.20 equiv) and DIPEA (219 µL, 1.35 mmol, 1.20 equiv) were added. The reaction was stirred at 30 °C for 4 h and the crude product was purified by flash column chromatography on silica gel using hexane:ethyl acetate (5:5) to afford **16** (621 mg, 81%) as a pale-yellow oil. ^1^H NMR (300 MHz, CDCl_3_) *δ*: 8.50 (s, 1H, NH-1^i^Trp), 8.27 (t, *J* = 1.5 Hz, 1H Ar), 8.24 (d, *J* = 1.5 Hz, 2H, Ar), 7.43 (d, *J* = 7.0 Hz, 1H, Ar), 7.28 (m, 1H, Ar), 7.06 (m, 2H, Ar), 5.70 (bs, 1H, NHCO), 5.15 (d, *J* = 8.0 Hz, 1H, NHCO), 4.52 (m, 1H, α-CHTrp), 3.84 (s, 6H, OCH_3_), 3.64 (s, 3H, OCH_3_), 3.20 (m, 4H, *β*-CH_2_Trp + CH_2_), 2.66 (t, *J* = 7.7 Hz, 2H, CH_2_), 1.54 (m, 2H, CH_2_), 1.38 (s, 8H, CH_2_), 1.21 (m, 11H, CH_3_ + CH_2_). ^13^C NMR (75 MHz, CDCl_3_) *δ*: 173.2, 166.5, 156.0, 155.6, 140.4, 137.7, 135.6, 131.2, 128.7, 124.4, 123.9, 121.3, 119.4, 118.0, 110.7, 105.2, 80.2, 54.7, 52.5, 52.5, 40.1, 29.9, 29.6, 29.1, 28.9, 28.4, 28.1, 27.4, 26.7, 26.0. HPLC (gradient: H_2_O:MeCN, 50–100% of MeCN in 10 min): 6.439 min.

#### 3.1.14. Tetramer **20**


To a cold (0 °C) solution of compound **14** (150 mg, 0.25 mmol) in dichloromethane (8 mL), TFA (0.4 mL) was added. The mixture was stirred at room temperature for 5 h, then volatiles were evaporated to dryness and the residue was coevaporated successively with methanol and dichloromethane. The residue (compound **17**) (100 mg, 0.20 mmol, 4.80 equiv) was next added to a mixture of **4 [27]** (18 mg, 0.04 mmol, 1.00 equiv), HATU (77 mg, 0.20 mmol, 4.80 equiv), and DIPEA (68 µL, 0.42 mmol, 10.00 equiv). The mixture was treated as described in the general coupling procedure. Purification by Biotage HPFC (high-performance flash chromatography) system on reverse phase using water:acetonitrile (100:0 to 0:100) yielded **20** (21 mg, 20%). ^1^H NMR (500 MHz, DMSO-*d*_6_) *δ*: 10.78 (s, 4H, NH-1^i^Trp), 9.08 (s, 4H, NHCO), 8.34 (d, *J* = 7.8 Hz, 4H, Ar), 8.28 (m, 8H, Ar), 8.00 (t, *J* = 1.6 Hz, 4H, Ar), 7.39 (d, *J* = 7.8 Hz, 4H, Ar), 7.22 (d, *J* = 7.8 Hz, 4H, Ar), 6.96 (t, *J* = 7.5 Hz, 4H, Ar), 6.90 (t, *J* = 7.5 Hz, 4H, Ar), 6.38 (t, *J* = 5.9 Hz, 4H, NHCO), 4.50 (m, 4H, α-CHTrp), 3.85 (s, 24H, OCH_3_), 3.46 (m, 20H, OCH_3_ + CH_2_), 3.10 (m, 20H, *β*-CH_2_Trp + CH_2_), 2.97 (dd, *J* = 14.4, 6.9 Hz, 4H, *β*-CH_2_Trp), 2.68 (m, 8H, CH_2_), 2.33 (m, 4H, CH_2_), 2.23 (m, 4H, CH_2_), 1.80 (quint, *J* = 7.6 Hz, 8H, CH_2_). ^13^C NMR (126 MHz, DMSO-*d*_6_) *δ*: 172.4, 170.3, 165.5, 155.0, 141.7, 136.8, 135.4, 130.5, 128.0, 122.1, 121.8, 120.2, 118.2, 117.4, 110.6, 105.2, 68.8, 66.9, 53.3, 52.4, 51.7, 35.6, 30.0, 26.7, 22.9. HPLC (gradient: H_2_O:MeCN, 10–100% of MeCN in 10 min): 10.357 min.

#### 3.1.15. Tetramer **21**


To a cold (0 °C) solution of compound **15** (130 mg, 0.20 mmol) in dichloromethane (8 mL), TFA (0.4 mL) was added. The mixture was stirred at room temperature for 5 h, then volatiles were evaporated to dryness and the residue was coevaporated successively with methanol and dichloromethane. The residue (compound **18**) (110 mg, 0.17 mmol, 4.80 equiv) was next added to a mixture of **4 [27]** (15 mg, 0.04 mmol, 1.00 equiv), HATU (64 mg, 0.17 mmol, 4.80 equiv), and DIPEA (57 µL, 0.35 mmol, 10.00 equiv). The mixture was treated as described in the general coupling procedure. Purification by Biotage HPFC (high-performance flash chromatography) system on reverse phase using water:acetonitrile (100:0 to 0:100) yielded **21** (64 mg, 73%). ^1^H NMR (300 MHz, MeOD-*d*_4_) *δ*: 10.16 (s, 4H, NH-1^i^Trp), 8.22 (d, *J* = 1.5 Hz, 8H, Ar), 8.14 (t, *J* = 1.5 Hz, 4H, Ar), 7.38 (d, *J* = 6.9 Hz, 4H, Ar), 7.19 (d, *J* = 6.9 Hz, 4H, Ar), 6.93 (m, 8H, Ar), 6.16 (t, *J* = 5.7 Hz, 4H, NHCO), 4.70 (m, 4H, *α*-CHTrp), 3.87 (s, 24H, OCH_3_), 3.52 (m, 12H, OCH_3_), 3.32 (m, 16H, CH_2_), 3.18 (m, 12H, *β*-CH_2_Trp + CH_2_) 3.02 (m, 12H, CH_2_), 2.70 (qd, *J* = 7.6, 4.1 Hz, 8H, CH_2_), 2.31 (quint, *J* = 5.6 Hz, 8H, CH_2_), 1.71 (m, 8H, CH_2_), 1.53 (m, 8H, CH_2_), 1.39 (m, 8H, CH_2_). ^13^C NMR (126 MHz, CDCl_3_) *δ*: 166.4, 156.6, 140.4, 137.8, 135.9, 131.2, 128.2, 124.4, 124.1, 121.0, 119.1, 117.6, 111.2, 104.6, 52.7, 52.5, 44.4, 40.0, 38.1, 29.7, 29.5, 27.0, 26.2, 7.8. HPLC (gradient: H_2_O:MeCN, 60–100% of MeCN in 10 min): 5.615 min. HRMS (ESI^+^) *m/z*: calculated for C_128_H_151_N_16_O_36_ 2488.04769; found 2488.04916.

#### 3.1.16. Tetramer **22**

To a cold (0 °C) solution of compound **16** (400 mg, 0.59 mmol) in dichloromethane (10 mL), TFA (0.5 mL) was added. The mixture was stirred at room temperature for 5 h, then volatiles were evaporated to dryness and the residue was coevaporated successively with methanol and dichloromethane. The residue (compound **19**) (350 mg, 0.50 mmol, 4.80 equiv), was next added to a mixture of **4 [27]** (44 mg, 0.10 mmol, 1.00 equiv), HATU (191 mg, 0.50 mmol, 4.80 equiv), and DIPEA (126 µL, 1.05 mmol, 10.00 equiv). The mixture was treated as described in the general coupling procedure. Purification by Biotage HPFC (high-performance flash chromatography) system on reverse phase using water:acetonitrile (100:0 to 0:100) yielded **22** (219 mg, 78%). ^1^H NMR (500 MHz, DMSO-*d*_6_) *δ*: 10.69 (s, 4H, NHCO), 8.97 (s, 4H, NH-1^i^Trp), 8.31 (d, *J* = 7.7 Hz, 4H, NHCO), 8.27 (d, *J* = 1.6 Hz, 8H, Ar), 8.00 (t, *J* = 1.6 Hz, 4H, Ar), 7.38 (d, *J* = 7.8 Hz, 4H, Ar), 7.20 (d, *J* = 7.8 Hz, 4H, Ar), 6.95 (t, *J* = 7.3 Hz, 4H, Ar), 6.89 (t, *J* = 7.3 Hz, 4H, Ar), 6.21 (t, *J* = 5.6 Hz, 4H, NHCO), 4.50 (m, 4H, *α*-CHTrp), 3.86 (s, 24H, OCH_3_), 3.46 (s, 12H, OCH_3_), 3.41 (m, 4H, CH_2_), 3.12 (m, 8H, CH_2_), 3.07 (m, 12H, *β*-CH_2_Trp + CH_2_), 2.95 (dd, *J* = 14.3, 6.8 Hz, 4H, *β*-CH_2_Trp), 2.63 (m, 8H, CH_2_), 2.31 (m, 4H, CH_2_), 2.23 (m, 4H, CH_2_), 1.59 (m, 8H, CH_2_), 1.41 (m, 8H, CH_2_), 1.27 (m, 36H, CH_2_). ^13^C NMR (126 MHz, DMSO-*d*_6_) *δ*: 172.4, 170.2, 165.5, 154.9, 141.7, 137.4, 135.3, 130.5, 128.0, 122.0, 121.8, 120.0, 118.1, 117.3, 110.5, 105.0, 68.7, 66.9, 53.3, 52.4, 51.6, 44.9, 35.6, 29.7, 29.3, 28.9, 28.9 28.8, 26.8, 26.4, 25.3. HPLC (gradient: H_2_O:MeCN, 10–100% of MeCN in 10 min): 11.256 min. HRMS (ESI^+^): calculated for C_140_H_175_N_16_O_36_ 2656.23549; found 2656.24812. 

#### 3.1.17. Tetramer **23**

According to the general procedure for deprotection of methyl esters, compound **20** (20 mg, 0.01 mmol, 1.00 equiv) was treated with LiOH·H_2_O (9 mg, 0.20 mmol, 24.00 equiv). Purification by Biotage HPFC (high-performance flash chromatography) system on reverse phase using water:acetonitrile (100:0 to 0:100) yielded **23** (15 mg, 79%). ^1^H NMR (500 MHz, DMSO-*d*_6_) *δ*: 10.72 (s, 4H, NH-1^i^Trp), 8.29 (s, 8H, Ar), 8.15 (s, 4H, NHCO), 8.01 (s, 4H, Ar), 7.51 (d, *J* = 7.7 Hz, 4H, Ar), 7.18 (d, *J* = 7.7 Hz, 4H, Ar), 6.93 (t, *J* = 7.5 Hz, 4H, Ar), 6.88 (t, *J* = 7.5 Hz, 4H, Ar), 4.48 (m, 4H, *α*-CHTrp), 3.32 (m, 8H, CH_2_), 3.13 (m, 12H, *β*-CH_2_Trp + CH_2_), 3.03 (s, 8H, CH_2_), 2.95 (m, 4H, *β*-CH_2_Trp), 2.73 (t, *J* = 7.4 Hz, 8H, CH_2_), 2.30 (dt, *J* = 14.2, 6.5 Hz, 4H, CH_2_), 2.17 (dt, *J* = 14.2, 6.5 Hz, 4H, CH_2_), 1.79 (m, 8H, CH_2_). ^13^C NMR (126 MHz, DMSO-*d*_6_) *δ*: 170.0, 167.3, 155.5, 141.6, 136.7, 135.3, 128.4, 122.2, 121.8, 119.9, 118.1, 117.9, 110.4, 106.4, 68.7, 66.8, 64.9, 54.2, 44.7, 35.9, 29.8, 22.9, 15.2. HPLC (gradient: H_2_O:MeCN, 10–100% of MeCN in 10 min): 6.997 min. HRMS (ESI^+^) *m/z*: calculated for C_109_H_116_N_16_O_36_ 2224.77381; found 2224.76703. 

#### 3.1.18. Tetramer **24**


According to the general procedure for deprotection of methyl esters, compound **21** (63 mg, 0.02 mmol, 1.00 equiv) was treated with LiOH·H_2_O (25 mg, 0.60 mmol, 24.00 equiv). Purification by Biotage HPFC (high-performance flash chromatography) system on reverse phase using water:acetonitrile (100:0 to 0:100) yielded **24** (20 mg, 34%). ^1^H NMR (500 MHz, DMSO-*d*_6_) *δ*: 10.65 (s, 4H, NHCO), 9.33 (s, 4H, NH-1^i^Trp), 8.23 (s, 8H, Ar), 8.11 (d, *J* = 8.2 Hz, 4H, NHCO), 8.00 (s, 4H, Ar), 7.46 (d, *J* = 7.8 Hz, 4H, Ar), 7.18 (d, *J* = 7.8 Hz, 4H, Ar), 6.92 (t, *J* = 7.4 Hz, 4H, Ar), 6.87 (t, *J* = 7.4 Hz, 4H, Ar), 6.72 (m, 4H, NHCO), 4.47 (m, 4H, *α*-CHTrp), 3.32 (m, 8H, CH_2_), 3.09 (m, 20H, *β*-CH_2_Trp + CH_2_), 2.94 (dd, *J* = 14.3, 6.9 Hz, 4H, *β*-CH_2_Trp), 2.66 (m, 8H, CH_2_), 2.28 (m, 4H, CH_2_), 2.20 (m, 4H, CH_2_), 1.63 (quint, *J* = 7.3 Hz, 8H), 1.47 (quint, *J* = 7.3 Hz, 8H, CH_2_), 1.33 (quint, *J* = 7.3, 8H, CH_2_). ^13^C NMR (126 MHz, DMSO-*d*_6_) *δ*: 174.0, 170.0, 167.2, 155.2, 141.4, 137.4, 135.3, 132.1, 128.4, 122.3, 121.9, 119.9, 118.0, 117.8, 110.4, 105.9, 68.8, 66.9, 53.8, 44.7, 40.4, 35.8, 29.4, 29.1, 27.1, 26.4, 25.4. HPLC (gradient: H_2_O:MeCN, 10–100% of MeCN in 10 min): 7.043 min. 

#### 3.1.19. Tetramer **25**

According to the general procedure for deprotection of methyl esters, compound **22** (117 mg, 0.04 mmol, 1.00 equiv) was treated with LiOH·H_2_O (44 mg, 1.06 mmol, 24.00 equiv). Purification by Biotage HPFC (high-performance flash chromatography) system on reverse phase using water/acetonitrile (100:0 to 0:100) yielded **25** (88 mg, 80%). ^1^H NMR (500 MHz, DMSO-*d*_6_) *δ*: 8.99 (s, 4H, NH-1^i^Trp), 8.22 (s, 8H, Ar), 8.13 (d, *J* = 8.1 Hz, 4H, NHCO), 8.01 (s, 4H, Ar), 7.45 (d, *J* = 7.7 Hz, 4H, Ar), 7.19 (d, *J* = 7.9 Hz, 4H, Ar), 6.94 (t, *J* = 7.5 Hz, 4H, Ar), 6.88 (t, *J* = 7.5 Hz, 4H, Ar), 6.33 (s, 4H, NHCO), 4.46 (m, 4H, *α*-CHTrp), 3.38 (m, 8H), 3.08 (s, 16H, *β*-CH_2_Trp + CH_2_), 2.93 (s, 4H, *β*-CH_2_Trp), 2.65 (m, 8H, CH_2_), 2.29 (m, 4H, CH_2_), 2.21 (m, 4H, CH_2_), 1.61 (m, 8H, CH_2_), 1.41 (m, 8H, CH_2_), 1.27 (m, 36H, CH_2_). ^13^C NMR (126 MHz, DMSO-*d_6_*) *δ*: 173.6, 170.1, 166.9, 155.0, 141.3, 137.5, 135.3, 131.8, 128.3, 122.4, 122.0, 119.9, 118.1, 117.7, 110.4, 105.6, 68.8, 66.9, 53.5, 44.8, 35.8, 29.7, 29.4, 28.9, 28.8, 26.5, 25.5. HPLC (gradient: H_2_O:MeCN, 10–100% of MeCN in 10 min): 7.876 min. HRMS (ESI^-^) *m/z*: calculated for C_129_H_156_N_16_O_36_ 2505.08682; found 2505.08471. 

#### 3.1.20. Methyl 4-(3-{2-[(*tert*-Butoxycarbonyl)amino]-3-methoxy-3-oxopropyl}-1*H*-indol-2-yl)butanoate **26**


According to the general C2 alkylation procedure, *N^α^*-Boc-(*S*)-tryptophan methyl ester **7** (100 mg, 0.31 mmol, 1.00 equiv) was treated with norbornene (59 mg, 0.63 mmol, 2.00 equiv), K_2_CO_3_ (177 mg, 1.26 mmol, 4.00 equiv), PdCl_2_ (6 mg, 0.03 mmol, 10 mol%), and methyl 4-bromobutyrate (227 mg, 1.26 mmol, 4.00 equiv). Purification by CCTLC using dichloromethane:methanol (20:1) yielded **26** (126 mg, 96%) as a pale-yellow oil. ^1^H NMR (300 MHz, Acetone-*d*_6_) *δ*: 10.05 (s, 1H, NH-1^i^Trp), 7.49 (d, *J* = 7.9 Hz, 1H, Ar), 7.28 (d, *J* = 7.3 Hz, 1H, Ar), 7.00 (m, 2H, Ar), 6.00 (d, *J* = 8.5 Hz, 1H, NHCO), 4.43 (m, *J* = 7.3 Hz, 1H, α-CHTrp), 3.61 (m, 6H, OCH_3_), 3.23 (dd, *J* = 14.6, 7.1 Hz, 1H, *β*-CH_2_Trp), 3.12 (dd, *J* = 14.6, 7.1 Hz, 1H, *β*-CH_2_Trp), 2.83 (t, *J* = 7.6 Hz, 2H, CH_2_), 2.38 (t, *J* = 7.6 Hz, 2H, CH_2_), 1.95 (m, 2H, CH_2_), 1.34 (s, 9H, CH_3_). ^13^C NMR (101 MHz, Acetone-*d*_6_) *δ*: 173.9, 173.6, 156.0, 137.3, 136.8, 129.5, 121.5, 119.5, 118.7, 111.4, 107.1, 79.3, 55.5, 52.2, 51.6, 44.4, 35.6, 33.8, 28.5, 27.7, 25.8, 24.9. HPLC (gradient: H_2_O:MeCN, 10–100% of MeCN in 10 min): 8.829 min. HRMS (ESI^+^) *m/z*: calculated for C_22_H_30_N_2_O_6_ 418.21039; found 418.21162.

#### 3.1.21. Ethyl 7-(3-{2-[*(tert*-Butoxycarbonyl)amino]-3-methoxy-3-oxopropyl}-1*H*-indol-2-yl)heptanoate **27**


According to the general C2 alkylation procedure, *N^α^*-Boc-(*S*)-tryptophan ethyl ester **7** (800 mg, 2.51 mmol, 1.00 equiv) was treated with norbornene (473 mg, 5.03 mmol, 2.00 equiv), K_2_CO_3_ (1.78 g, 10.05 mmol, 4.00 equiv), PdCl_2_ (35 mg, 0.25 mmol, 10 mol%), and ethyl 7-bromoheptanoate (1.96 mL, 10.05 mmol, 4.00 equiv). Purification by flash column chromatography using dichloromethane:methanol (20:1) yielded **27** (872 mg, 77%) as a pale-yellow oil. ^1^H NMR (300 MHz, CDCl_3_) *δ*: 8.02 (s, 1H, NH-1^i^Trp), 7.44 (d, *J* = 7.5 Hz, 1H, Ar), 7.26 (m, 1H, Ar), 7.08 (m, 2H, Ar), 5.06 (d, *J* = 8.3 Hz, 1H, NHCO), 4.57 (m, 1H, α-CHTrp), 3.63 (s, 3H, OCH_3_), 3.21 (d, *J* = 5.7 Hz, 2H, *β*-CH_2_Trp), 2.70 (t, *J* = 7.7 Hz, 2H, CH_2_), 2.29 (t, *J* = 7.4 Hz, 2H, CH_2_), 1.65 (m, 6H, CH_2_), 1.38 (m, 13H, CH_3_ + CH_2_). ^13^C NMR (75 MHz, CDCl_3_) *δ*: 174.0, 173.0, 155.2, 137.3, 135.4, 128.8, 121.4, 119.5, 118.3, 110.5, 105.7, 60.4, 54.3, 52.4, 34.3, 29.6, 29.1, 28.9, 28.5, 27.5, 26.0, 24.9, 14.4. HPLC (gradient: H_2_O:MeCN, 60–100% of MeCN in 10 min): 4.706 min. HRMS (ESI^+^) *m/z*: calculated for C_26_H_38_N_2_O_6_ 474.27299; found 474.27158.

#### 3.1.22. Tetramer **30**

To a cold (0 °C) solution of compound **26** (169 mg, 0.40 mmol) in dichloromethane (15 mL), TFA (1 mL) was added. The mixture was stirred at room temperature for 5 h, then volatiles were evaporated to dryness and the residue (compound **28**) was coevaporated successively with methanol and dichloromethane. Intermediate **28** (150 mg, 0.35 mmol, 4.80 equiv) was added under argon atmosphere to a solution of **4 [27]** (31 mg, 0.07 mmol, 1.00 equiv), HATU (132 mg, 0.35 mmol, 4.80 equiv), and DIPEA (117 µL, 0.72 mmol, 10.00 equiv) in anhydrous DMF (15 mL). The reaction mixture was treated as described in the general coupling procedure. Purification by Biotage HPFC (high-performance flash chromatography) system on reverse phase using water:acetonitrile (100:0 to 0:100) yielded **30** (32 mg, 26%).^1^H NMR (500 MHz, CDCl_3_) *δ*: 9.06 (s, 4H, NH-1^i^Trp), 7.39 (m, 4H, Ar), 7.10 (m, 4H, Ar), 7.00 (m, 8H, Ar), 6.77 (d, *J* = 8.2 Hz, 4H, NHCO), 4.90 (m, 4H, *α*-CHTrp), 3.65 (s, 24H, OCH_3_), 3.26 (dd, *J* = 14.8, 5.6 Hz, 4H, *β*-CH_2_Trp), 3.19 (dd, *J* = 14.8, 5.6 Hz, 4H, *β*-CH_2_Trp), 3.06 (m, 4H, OCH_2_), 3.02 (m, 4H, OCH_2_), 2.70 (m, 10H, CH_2_), 2.45 (d, *J* = 9.3 Hz, 6H, CH_2_), 2.30 (t, *J* = 7.2 Hz, 8H, CH_2_), 2.20 (m, 8H, CH_2_), 1.95 (quint, *J* = 7.2 Hz, 8H, CH_2_). ^13^C NMR (126 MHz, CDCl_3_) *δ*: 174.1, 172.7, 172.2, 136.3, 135.7, 128.6, 121.4, 119.4, 117.9, 111.0, 105.7, 70.0, 66.6, 53.2, 52.5, 51.8, 51.8, 44.1, 36.6, 33.1, 26.8, 25.2, 25.0. HPLC (gradient: H_2_O:MeCN, 30–95% of MeCN in 10 min): 9.230 min. 

#### 3.1.23. Tetramer **31**

To a cold (0 °C) solution of compound **27** (850 mg, 1.79 mmol) in dichloromethane (15 mL), TFA (1 mL) was added. The mixture was stirred at room temperature for 5 h, then volatiles were evaporated to dryness and the residue (compound **29**) was coevaporated successively with methanol and dichloromethane. Intermediate **29** (800 mg, 1.64 mmol, 4.80 equiv) was added under argon atmosphere to a solution of **4 [27]** (144 mg, 0.34 mmol, 1.00 equiv), HATU (622 mg, 1.64 mmol, 4.80 equiv), and DIPEA (554 µL, 3.41 mmol, 10.00 equiv) in anhydrous DMF (15 mL) and treated as described in the general coupling procedure. Purification by Biotage HPFC (high-performance flash chromatography) system on reverse phase using water:acetonitrile (100:0 to 0:100) yielded **31** (304 mg, 48%). ^1^H NMR (400 MHz, CDCl_3_) *δ*: 8.90 (s, 4H, NH-1^i^Trp), 7.37 (d, *J* = 6.6, 4H, Ar), 7.13 (d, *J* = 6.6, 4H, Ar), 7.00 (m, 8H, Ar), 6.73 (d, *J* = 8.0 Hz, 4H, NHCO), 4.90 (m, 4H, *α*-CHTrp), 4.11 (q, *J* = 7.1 Hz, 8H, OCH_2_CH_3_), 3.63 (s, 12H, CH_3_), 3.25 (d, *J* = 5.5 Hz, 8H, *β*-CH_2_Trp), 3.06 (m, 8H, OCH_2_), 2.74 (d, *J* = 9.3 Hz, 4H, CH_2_), 2.64 (t, *J* = 7.5 Hz, 8H, CH_2_), 2.54 (d, *J* = 9.2 Hz, 4H, CH_2_), 2.26 (m, 16H, CH_2_), 1.59 (m, 16H, CH_2_), 1.30 (m, 16H, CH_2_), 1.24 (t, *J* = 7.1 Hz, 12H, OCH_2_CH_3_). ^13^C NMR (101 MHz, CDCl_3_) *δ*: 174.0, 172.7 172.1, 137.6, 135.6, 128.8, 121.2, 119.3, 117.8, 110.9, 105.0, 70.0, 66.6, 60.4, 53.2, 52.5, 44.2, 36.7, 34.3, 29.5, 29.0, 29.0, 26.9, 26.2, 24.9, 14.4. HPLC (gradient: H_2_O:MeCN, 60–100% of MeCN in 10 min): 8.092 min. HRMS (ESI^+^) *m/z*: calculated for C_101_H_140_N_8_O_24_ 1849.00616; found 1848.99805.

#### 3.1.24. Tetramer **32**

According to the general procedure for deprotection of methyl esters, compound **30** (56 mg, 0.04 mmol, 1.00 equiv) was treated with LiOH·H_2_O (24 mg, 0.57 mmol, 16.00 equiv). Purification by Biotage HPFC (high-performance flash chromatography) system on reverse phase using water:acetonitrile (100:0 to 0:100) yielded **32** (33 mg, 66%). ^1^H NMR (500 MHz, DMSO-*d*_6_) *δ*: 10.70 (s, 4H, NH-1^i^Trp), 8.17 (m, 4H, NHCO), 7.47 (d, *J* = 7.8 Hz, 4H, Ar), 7.20 (d, *J* = 7.8 Hz, 4H, Ar), 6.96 (t, *J* = 7.5 Hz, 4H, Ar), 6.90 (t, *J* = 7.5 Hz, 4H, Ar), 4.45 (m, 4H, *α*-CHTrp), 3.46–3.29 (m, 8H, CH_2_), 3.12–3.05 (m, 12H, *β*-CH_2_Trp + CH_2_), 2.92 (dd, *J* = 14.3, 7.3 Hz, 4H, *β*-CH_2_Trp), 2.70 (h, *J* = 7.1 Hz, 8H, CH_2_), 2.35–2.25 (m, 4H, CH_2_), 2.25–2.14 (m, 12H, CH_2_), 1.86 (quint, *J* = 7.3 Hz, 8H, CH_2_). ^13^C NMR (126 MHz, DMSO-*d*_6_) *δ*: 174.4, 173.8, 170.1, 136.6, 135.4, 128.2, 120.1, 118.1, 117.8, 110.5, 106.3, 68.9, 67.0, 53.7, 44.7, 35.8, 33.4, 26.9, 24.8. HPLC (gradient: H_2_O:MeCN, 10–100% of MeCN, in 10 min): 7.123 min. HRMS (ESI^+^) *m/z*: calculated for C_77_H_92_N_8_O_24_ 1512.62245; found 1512.62337. 

#### 3.1.25. Tetramer **33**

According to the general procedure for deprotection of methyl esters, compound **31** (97 mg, 0.05 mmol, 1.00 equiv) was treated with LiOH·H_2_O (35 mg, 0.84 mmol, 16.00 equiv). Purification by Biotage HPFC (high-performance flash chromatography) system on reverse phase using water:acetonitrile (100:0 to 0:100) yielded **33** (25 mg, 28%). ^1^H NMR (500 MHz, DMSO-*d*_6_) *δ*: 10.65 (s, 4H, NH-1^i^Trp), 8.17 (d, *J* = 8.0 Hz, 4H, NHCO), 7.46 (d, *J* = 7.8 Hz, 4H, Ar), 7.19 (d, *J* = 7.8 Hz, 4H, Ar), 6.94 (t, *J* = 7.5 Hz, 4H, Ar), 6.89 (t, *J* = 7.5 Hz, 4H, Ar), 4.45 (m, 4H, *α*-CHTrp), 3.39 (m, 16H, CH_2_), 3.10 (m, 12H, *β*-CH_2_Trp + OCH_2_), 2.93 (dd, *J* = 14.3, 7.0 Hz, 4H, *β*-CH_2_Trp), 2.63 (m, 8H, CH_2_), 2.18 (t, *J* = 7.4 Hz, 8H, CH_2_), 1.61 (q, *J* = 7.1 Hz, 8H, CH_2_), 1.48 (m, 8H, CH_2_), 1.30 (m, 16H, CH_2_). ^13^C NMR (126 MHz, DMSO-*d*_6_) *δ*: 174.5, 173.7, 170.0, 137.4, 135.3, 128.3, 119.9, 118.0, 117.7, 110.4, 105.7, 68.9, 67.0, 53.6, 44.7, 35.8, 33.7, 29.2, 28.7, 28.4, 25.4, 24.5. HPLC (gradient: H_2_O:MeCN, 10–100% of MeCN in 10 min): 7.873 min. HRMS (ESI^+^) *m/z*: calculated for C_89_H_116_N_8_O_24_ 1680.81025; found 1680.80782.

#### 3.1.26. Tetramer **34**

Commercially available dimethyl 5-aminobenzene-1,3-dicarboxylate (236 mg, 1.13 mmol, 4.80 equiv) was added under argon atmosphere to a mixture of **4 [27]** (100 mg, 0.24 mmol, 1.00 equiv), HATU (430 mg, 1.13 mmol, 4.80 equiv), and DIPEA (383 µL, 2.36 mmol, 10.00 equiv) in anhydrous DMF (15 mL). The reaction mixture was treated as described in the general coupling procedure. The crude product was purified by CCTLC using hexane:ethyl acetate (1:9) to afford **34** (211 mg, 75%). ^1^H NMR (300 MHz, CDCl_3_) *δ*: 9.00 (s, 4H, NHCO), 8.42 (d, *J* = 1.6 Hz, 8H, Ar), 8.34 (t, *J* = 1.4 Hz, 4H, Ar), 3.90 (s, 24H, OCH_3_), 3.74 (t, *J* = 5.5 Hz, 8H, OCH_2_CH_2_), 3.42 (s, 8H, CH_2_O), 2.61 (t, *J* = 5.5 Hz, 8H, OCH_2_CH_2_). ^13^C NMR (126 MHz, DMSO-*d_6_*) *δ*: 170.8, 166.2, 139.0, 131.3, 126.2, 125.2, 69.6, 67.4, 52.7, 45.6, 37.9. HPLC (gradient: H_2_O:MeCN, 10–100% of MeCN in 10 min): 9.029 min. 

#### 3.1.27. Tetramer **35**

According to the general procedure for deprotection of methyl esters, compound **34** (125 mg, 0.11 mmol, 1.00 equiv) was treated with LiOH·H_2_O (71 mg, 1.68 mmol, 16.00 equiv). Purification by Biotage HPFC (high-performance flash chromatography) system on reverse phase using water/acetonitrile (100:0 to 0:100) yielded **35** (89 mg, 78%).^1^H NMR (500 MHz, DMSO-*d_6_*) *δ*: 10.23 (s, 4H, NHCO), 8.41 (d, *J* = 1.6 Hz, 8H, Ar), 8.12 (d, *J* = 1.6 Hz, 4H, Ar), 3.55 (t, *J* = 6.3 Hz, 8H, OCH_2_CH_2_), 3.27 (s, 8H, CH_2_O), 2.46 (t, *J* = 6.3 Hz, 8H, OCH_2_CH_2_). ^13^C NMR (126 MHz, DMSO-*d_6_*) *δ*: 169.7, 166.5, 139.7, 131.7, 124.5, 123.5, 69.1, 67.1, 45.2, 37.2. HPLC (gradient: H_2_O:MeCN, 10–100% of MeCN in 10 min): 4.989 min. HRMS (ESI^-^) *m/z*: calculated for C_49_H_48_N_4_O_24_ 1076.26585; found 1076.26573.

#### 3.1.28. Dimethyl 5-[3-(2-{[(Benzyloxy)carbonyl]amino}ethyl)ureido]isophthalate **37**


A solution of *N*-Cbz-ethylenediamine hydrochloride **36** (500 mg, 2.17 mmol, 1.00 equiv) in anhydrous DMF (5 mL) was treated with dimethyl 5-isocyanatoisophthalate (765 mg, 3.25 mmol, 1.50 equiv) and DIPEA (1.13 mL, 6.50 mmol, 3.00 equiv). The reaction was stirred at 30 °C overnight under argon atmosphere and the crude product was purified by flash column chromatography on silica gel using hexane:ethyl acetate (7:3) to afford **37** (500 mg, 54%) as a white amorphous solid. ^1^H NMR (400 MHz, DMSO-*d*_6_) *δ*: 9.13 (s, 1H, Ar), 8.30 (d, *J* = 1.4 Hz, 2H, Ar), 8.02 (s, 1H, NHCO), 7.42–7.20 (m, 6H, Ar + NHCO), 6.30 (t, *J* = 5.7 Hz, 1H, NHCO), 5.02 (s, 2H, -CH_2_Ph), 3.87 (s, 6H, OCH_3_), 3.22–3.15 (m, 2H, CH_2_), 3.15–3.07 (m, 2H, CH_2_). HPLC (gradient: H_2_O:MeCN, 10–100% of MeCN in 10 min): 7.609 min. 

#### 3.1.29. Dimethyl 5-[3-(2-Aminoethyl)ureido]isophthalate **38**


A solution of **37** (264 mg, 0.61 mmol, 1.00 equiv) in DMF (10 mL) was treated with ammonium formate (128 mg, 2.03 mmol, 3.30 equiv) and Pd/C (10% on C; 30 wt %) under argon atmosphere. After 3 h, the residue was filtered through a Whatman^®^ filter paper 42 and the solvent was removed under reduced pressure to give 137 mg of **38** as a brown oil. The crude was used for the next step without purification. ^1^H NMR (300 MHz, CDCl_3_) *δ*: 8.08 (d, *J* = 1.6 Hz, 2H, Ar), 8.05 (t, *J* = 1.5 Hz, 1H, Ar), 7.98 (m, NH), 6.52 (s, 2H, NH_2_), 3.79 (s, 6H, OCH_3_), 3.55–3.35 (m, 4H, CH_2_). HPLC (gradient: H_2_O:MeCN, 15–95% of MeCN in 10 min): 4.981 min. 

#### 3.1.30. Tetramer **39**


Compound **38** (130 mg, 0.44 mmol, 4.80 equiv) was added under argon atmosphere to a mixture of **4** [27] (39 mg, 0.09 mmol, 1.00 equiv), HATU (167 mg, 0.44 mmol, 4.80 equiv), and DIPEA (150 µL, 0.92 mmol, 10.00 equiv) in anhydrous DMF (10 mL). The reaction mixture was treated as described in the general coupling procedure. The crude product was purified by Biotage HPFC (high-performance flash chromatography) system on reverse phase using water:acetonitrile (100:0 to 0:100) to afford **39** (48 mg, 34%). ^1^H NMR (500 MHz, CD_3_OD) *δ*: 8.15 (d, *J* = 1.6 Hz, 8H, Ar), 8.07 (t, *J* = 1.6 Hz, 4H, Ar), 3.88 (s, 24H, OCH_3_), 3.58 (t, *J* = 6.0 Hz, 8H, -OCH_2_CH_2_CO-), 3.39–3.32 (m, 16H, CH_2_), 3.25 (s, 8H, CH_2_), 2.40 (*t*, *J* = 6.0 Hz, 8H, -OCH_2_CH_2_CO-). HPLC (gradient: H_2_O:MeCN, 10–100% of MeCN in 10 min): 7.866 min. 

#### 3.1.31. Tetramer **40**

According to the general procedure for deprotection of methyl esters, compound **39** (25 mg, 0.02 mmol, 1.00 equiv) was treated with LiOH·H_2_O (42 mg, 0.26 mmol, 16.00 equiv). Purification by Biotage HPFC (high-performance flash chromatography) system on reverse phase using water/acetonitrile (100:0 to 0:100) yielded product **40** (15 mg, quant.). ^1^H NMR (500 MHz, DMSO-*d_6_*) *δ*: 9.78 (s, 4H, NHPh), 8.25 (s, 8H, Ar), 8.04 (d, *J* = 1.6 Hz, 4H, Ar), 8.00 (d, *J* = 6.2 Hz, 3H, NHCO), 7.34 (s, 3H, NHCO), 3.45 (t, *J* = 6.4 Hz, 8H, CH_2_), 3.17 (s, 16H, CH_2_), 3.04 (m, 8H, CH_2_), 2.26 (q, *J* = 6.3 Hz, 8H, CH_2_). ^13^C NMR (126 MHz, DMSO-*d_6_*) *δ*: 170.6, 168.0, 155.6, 141.1, 133.3, 122.5, 121.6, 68.3, 67.2, 44.9, 36.3, 31.3, 29.0. HPLC (gradient: H_2_O:MeCN, 10–100% of MeCN in 10 min): 4.781 min. HRMS (ESI^+^) *m/z*: calculated for C_61_H_72_N_12_O_28_ 1420.4579; found 1420.45595.

### 3.2. Biological Methods 

#### 3.2.1. Antiviral Activity against HIV

The MT-4 cells used for the anti-HIV assays were a kind gift from Dr. L. Montagnier (formerly at the Pasteur Institute, Paris, France) and were cultured in RPMI-1640 medium (Invitrogen, Merelbeke, Belgium) supplemented with 10% fetal calf serum (Hyclone, Perbio Science, Aalst, Belgium) and 1% L-glutamine (Invitrogen). The HIV-1 strain NL4-3 was obtained from the AIDS Research and Reference Reagent Program (Division of AIDS, NIAID, NIH) and cultured in MT-4 cells. The virus stock was stored at −80 °C. 

The compounds were evaluated for their inhibitory activity against HIV-1 (NL4.3) and HIV-2 (ROD) infection in MT-4 cell cultures as described in detail earlier [39]. Briefly, MT-4 cells (50 µL, 1 × 106 cells/mL) were preincubated for 30 min at 37 °C with the test compounds (100 µL) in a 96-well plate. Next, the cell-line-adapted HIV strains (NL4.3 and ROD) were added according to the TCID50 (50% tissue culture infectious dose) of the viral stock. After 5 days, the cytopathic effect (CPE) was scored microscopically and the anti-HIV-1 activity (50% effective concentration, EC_50_) of each compound was calculated using a colorimetric method based on the in situ reduction of 3-(4,5-dimethylthiazol-2-yl)-5-(3-carboxymethoxyphenyl)-2-(4-sulfophenyl)-2*H*-tetrazolium (MTS) [40] according to the manufacturer’s instructions (Promega, Leiden, The Netherlands). Assays were performed by adding a small amount of a solution that contains MTS and an electron-coupling reagent (phenazine ethosulfate; PES) directly to culture wells, incubating for 1–4 h, and then recording absorbance at 490 nm (A490) with a 96-well plate reader. The quantity of formazan product, as measured by the magnitude of 490 nm absorbance, is directly proportional to the number of living cells in culture. Cytotoxicity in MT-4 cells was measured after 5 days using the MTS/PES method [39,40]. Data are the mean ± S.D. of at least three independent experiments.

#### 3.2.2. Antiviral Activity against EV-A71

The EV-A71 laboratory-adapted BrCr strain and clinical isolates representative of B genogroup (B2 subgenogroup: 11316; B5 subgenogroup: TW/70902/08) and C genogroup (C2 subgenogroup: H08300 461#812; C4 subgenogroup: TW/1956/05) were used at a low multiplicity of infection (MOI) in a standardized cell-based antiviral assay. Briefly, rhabdosarcoma (RD) cells were seeded in a 96-well plate. The day after, a serial dilution of the compounds and the virus inoculum were added to the cells. The assay plates were incubated at 37 °C, 5% CO_2_ with virus inoculum and compounds until full virus-induced cell death was observed in the untreated, infected controls (3 days post-infection). Subsequently, the antiviral effect was quantified using a colorimetric readout with MTS/phenazine methosulfate (MTS/PMS method), and the compound concentration at which 50% inhibition of virus-induced cell death is observed (EC_50_) was calculated from the antiviral dose–response curves. A similar assay setup was used to determine the adverse effect of the compound on uninfected, treated cells for calculation of the CC_50_ (concentration of compound that reduces overall cell health by 50% as determined with the MTS/PMS method). The selectivity index (SI) was calculated as the ratio of CC_50_ to EC_50_. 

### 3.3. Computer-Assisted Molecular Modeling

The protein model consisted of a pentamer of VP1 subunits extracted from the 3D structure of EV-A71 strain 11316 in complex with the Trp dendrimer **AL-385** (Protein Data Bank accession code 6DIZ) [23] upon replacement of Thr184 with Ser followed by energy minimization. Each VP1 subunit encompassed residues 78–280 that were conveniently “capped” by acetyl (ACE) and *N*-methyl amide (NME) groups at *N*- and *C*-termini, respectively. The molecular graphics program PyMOL [41] was used for model-building suitable **23 (AL-534)** fragments and assembling them into the full tetrapodal derivative, as well as for trajectory visualization and 3D figure generation. Geometry optimization and point charge derivation for a methyl-capped l-Trp residue with the urea-containing spacer covalently attached to the C2 position of the indole ring were achieved by means of the AM1-BCC model implemented in the *sqm* program [42]. The *ff14SB* [43] and *gaff* [44] force field parameter sets in AMBER 18 [45] were used for protein and ligand atoms, respectively. A three-dimensional cubic grid consisting of 65 × 65 × 65 points with a spacing of 0.375 Å centered on the Pro246 residue of one VP1 subunit was defined for docking purposes. AutoDock Vina 1.2 [46] was used to generate up to 10 feasible binding poses for each fragment studied, and the best poses were ranked according to intra- and intermolecular energy evaluations. 

The molecular dynamics simulations of the complexes formed between the VP1 pentamer and either the fragments or the full **23** (**AL-534**) molecule were run in explicit solvent for 300 ns at 300 K and 1 atm essentially as described before [22] using the *pmemd.cuda* engine [47], as implemented in AMBER 18. Briefly, each complex was immersed in a cubic box of TIP3P water molecules that extended 12 Å away from any ligand atom, and three chloride ions were added to achieve electroneutrality. The SHAKE algorithm and the smooth particle mesh Ewald method were employed. To avoid artifactual distortions due to the absence of neighboring capsid proteins, a weak harmonic restraint of 1.0 kcal·mol^−1^·Å^−2^ was applied on Cα atoms of VP1, except for those in loops B–C, D–E and H–I in each subunit. In addition, a simulated annealing procedure was followed to cool down (from 300 to 273 K over 1 ns) snapshots taken from the MD trajectory every 5 ns. The geometries of these “frozen” systems were then optimized by following an energy minimization protocol until the root mean square of the Cartesian elements of the gradient was less than 0.01 kcal·mol^−1^·Å^−1^. The final ensemble containing 30 energy-minimized frozen molecules, which can be expected to be closer to the global energy minimum, was taken as representative of the VP1-**23** (**AL-534**) complex. 

## 4. Conclusions

In this work, the influence of the stereochemistry of the Trp unit on the antiviral activity of selected tetrapodal derivatives was first studied by synthetizing **6**, the d-Trp isomer of **AL-471**. This compound displayed EC_50_ values against HIV and EV-A71 infection in the (sub)micromolar range similar to those found for **AL-471**. These results showed that the stereochemistry of the amino acid seems not to be crucial for antiviral activity. In addition, the importance of the indole ring of Trp for activity was more firmly established when we found that truncated compounds **35** and **40**, in which four isophthalic acid moieties were linked, directly or through a spacer, to the central scaffold, were inactive against HIV or considerably less potent than **AL-471** against EV-A71. In addition, replacement of the terminal isophthalic acid group with a nonaromatic carboxylic acid brought about decreased activity against both HIV and EV-A71; in fact, the C2 alkyl analogues **32** and **33**, were considerably less potent than the prototype **AL-471**.

We then confirmed that the distance between the isophthalic acid group attached to the C2 position of the indole moiety is indeed important for anti EV-A71 activity. When this moiety was separated from the C2 position by a 3-methylene linker (**23**, **AL-534**), the potency increased against EV-A71, whereas insertion of longer linkers made up of five and eight methylenes led to a loss of potency, as in **24** and **25**. Interestingly, we also found that **23** (**AL-534**) was more potent against all of the clinical EV-71 strains tested than the prototype **AL-471**. In fact, its subnanomolar potency against the clinical strains B5 and C4 was unprecedented in the series of reduced-sized compounds and was indeed comparable to that of the larger dendrimer **AL-385**.

The molecular modeling work presented here expands previous results from our group proposing that the decorated Trp units radiating from the pentaerythritol-based core of these tetrapodal derivatives direct the isophthalic acid moieties towards numerous pore-lining residues of the viral protein VP1 that are necessary for cell attachment. The novelty in the case of **23** (**AL-534**) is that the alkylurea linker that joins the indole ring of each l-Trp to the distal isophthalic acid presents an optimal length and suitable chemical properties that allow the establishment of unique interactions with amino acids that show sequence variation among EV-A71 subgenogroups.

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
