# Peer review of "Insertion of an Amphipathic Linker in a Tetrapodal Tryptophan Derivative Leads to a Novel and Highly Potent Entry Inhibitor of Enterovirus A71 Clinical Isolates"

_ijms, 2023, doi:10.3390/ijms24043539_

Round 1

Reviewer 1 Report

Dear Authors,

In the present work, the authors present relevant results regarding the synthesis of new antiviral compounds (dendrimers), making changes in the structure of a potent Enterovirus A71 inhibitor (AL-471) discovered by the group itself and published in previous works. The applied molecular simplification aimed to to assess the stereochemical preference of the Trp moiety for antiviral activity, insertion of flexible linker of different length in C2 and replace aromatic isophthalic acid residue and at the end of the C2 alkyl chain with a non-aromatic carboxylic acid. The most potent derivative (AL-534), with the C2 shortest alkyl urea linkage, showed subnanomolar potency against different EV-71 clinical isolates. The molecular modelling work presented expands previous results from research group proposing that the decorated Trp units radiating from the pentaerythritol-based core of these tetrapodal derivatives direct the isophthalic acid moieties towards numerous pore-lining residues of the viral protein VP1 that are necessary for cell attachment. The work is well written and the presented conclusions are consistent with the facts observed experimentally. Based on the above, I recommend the manuscript for publication in IJMS.

Author Response

This Reviewer considered that the work present relevant results, is well written and the presented conclusions are consistent with the facts observed experimentally. Based on that he/she recommended the manuscript for publication in IJMS

We are very grateful for the comments made by this reviewer.

Reviewer 2 Report

Dear authors,

His work entitled "INSERTION OF AN AMPHIPATHIC LINKER IN A TETRAPODAL TRYPTOPHAN DERIVATIVE LEADS TO A NOVEL AND HIGHLY POTENT ENTRY INHIBITOR OF ENTEROVIRUS A71 CLINICAL ISOLATES” is interesting and transcendent. However, some major and minor changes must be incorporated before publication.

I share some comments about it that I hope will be useful:

Major comments:

1. Results and discussion section – Table 1: Compounds 32 and 40 exhibit selectivity to HIV-2. I consider it necessary to add a brief paragraph that explains (hypothetically) the selectivity mechanism of these compounds.

2. Results and discussion section: Please provide the quality plots of the MD calculations (e.g. RMSD and RMSF values of the protein-ligand complex). These could be added as a part of the supplementary material. 

3. Results and discussion section: Is necessary to add the full results of in silico methods (e.g., docking binding scores, binding poses, and the trajectory obtained by the MD studied). I consider that your methodology is correctly structured, but it does not show concrete data that support the quality of your results. Is necessary an explicitly mention this data, and again, I recommend adding the supplementary material section to this data/figures/tables to support it.

4. Results and discussion section: I strongly recommend adding a final table with the drug-like properties calculated by the most representative compounds and their in vitro controls. This data could be very useful to the prospective optimization of these compounds. For example, to improve not only the activity profile of these compounds but parallelly also essential to know their possible pharmacological issues, like solubility, absorption, etc.  Also, I recommend adding a brief paragraph to explain the differences in drug-like properties between these compounds (if they exist it). I share with you a reference of free software (DataWarrior) that could be useful: https://doi.org/10.1080/17460441.2019.1581170

Minor comments:

1. Results and discussion section – Lines 197-198 and 236-237:  According to the sentence “This finding suggests that the stereochemistry of the amino acid is not crucial for the anti-HIV activity.” … Additionally, the selectivity index scores remarks on the importance of stereochemistry to understand the possible side effects generated by these compounds. I suggest expanding the discussion of these results.

2. Results and discussion section – Lines 297-299: According to the sentence “Finally, evaluation of 23 (AL-534) against single-mutant (VP1_S184T and VP1_P246S) and double-mutant (VP1_S184T_P246S) EV-A71 strains that are resistant to AL-385 showed diminished antiviral activity” … I suggest to say that this observation also is valid to Al-385 and AL-471.  Could  AL-471 interact in the same binding site as AL-385?

3. Materials and methods section – Lines 950: After the aa replacement, the protein-ligand complex was minimized? Not is clear, please add an explicit mention of this step.

4. Materials and methods section – Lines 961-962: What kind of charges were assigned? / What kind of searching algorithm was used to explore the conformational space?

5. Materials and methods section – Lines 967: What concentration of NaCl was used on this MD protocol? Or, not used?

Author Response

The response to reviewer  2 has been attached.

Round 2

Reviewer 2 Report

Dear authors,

My only observation about it is that the figures haven't a description, namely, the figures are not self-explicative. 

Regards.
